# The Risk Factors for Radiolucent Nephrolithiasis among Workers in High-Temperature Workplaces in the Steel Industry

**DOI:** 10.3390/ijerph192315720

**Published:** 2022-11-25

**Authors:** I-Cheng Lu, Chen-Cheng Yang, Chi-Hsien Huang, Szu-Ying Chen, Chi-Wei Lin, Chia-Hsiang Lin, Hung-Yi Chuang

**Affiliations:** 1Department of Occupational Medicine, E-Da Hospital, I-Shou University, Kaohsiung City 824, Taiwan; 2Department of Occupational Medicine, Kaohsiung Municipal Siaogang Hospital, Kaohsiung Medical University, Kaohsiung City 812, Taiwan; 3Department of Family Medicine, E-Da Hospital, I-Shou University, Kaohsiung City 824, Taiwan; 4Department of Urology Medicine, E-Da Hospital, I-Shou University, Kaohsiung City 824, Taiwan; 5Ph.D. Program in Environmental and Occupational Medicine, College of Medicine, and Research Center for Precision Environmental Medicine, Kaohsiung Medical University, Kaohsiung 807, Taiwan; 6Department of Occupational and Environmental Medicine, Kaohsiung Medical University Hospital, Kaohsiung City 807, Taiwan

**Keywords:** radiolucent stones, heat exposure, high-temperature workplace, steel worker

## Abstract

Workers in high-temperature workplaces with inadequate water supply may exhibit symptoms of chronic dehydration and have increased risk of nephrolithiasis. The aim of this study was to investigate the risk of radiolucent stone formation among workers in a high-temperature workplace and the related risk factors associated with the condition. We collected data from 1681 workers in a steel factory in Southern Taiwan who underwent regular health examinations. Radiolucent stones were defined as positive findings on ultrasound with negative radiographic images. The prevalences of nephrolithiasis and radiolucent stones in this study were 12.0% and 5.1%, respectively. Heat exposure and age were two major risk factors influencing the probability of radiolucent stones. We combined the age and heat exposure into four groups (over and under 35 years of age with and without heat exposure) in a logistic regression. For workers younger than 35 years, the odds ratio of radiolucent stones was 2.695 (95% confidence interval: 1.201–6.049) in workers with heat exposure compared to workers without. Our investigation further demonstrated that heat exposure was a main risk factor for radiolucent stone formation. In conclusion, our identification of heat exposure as an independent factor for radiolucent stone development in steel workers highlights the need for attention to be paid to those working in similar environments.

## 1. Introduction

The increasing prevalence of nephrolithiasis in the last decade has become a global problem [1,2,3,4,5,6,7]. The financial burden associated with urolithiasis has been estimated to be as high as EUR 54.38 million in Germany and up to USD 2 billion to 5.3 billion per year in the United States [8,9,10]. Nephrolithiasis not only increases the economic costs but also impairs the patients’ quality of life [11]. Nephrolithiasis is related to many factors, such as age, gender, water intake, hot climate, humidity, metabolic syndromes, and the heat environment [12,13,14,15,16,17,18].

Water intake is inversely related to the risk of kidney stones, and a large intake of water is essential for the therapy and prevention of stone recurrences [19,20]. In both men and women, an inverse association between fluid intake and urolithiasis risk in patients without a history of urolithiasis has been confirmed [21].

In addition, many studies have reported that metabolic syndrome, including obesity, hypertension, dyslipidemia, and hyperglycemia, increases the risk of nephrolithiasis [22,23,24,25]. Not only is body size positively associated with daily oxalate excretion [26], but obesity is also related to low urinary pH (potential of hydrogen), which may increase the risk of uric acid stone formation [27]. On the other hand, insulin resistance is noted to decrease urinary citrate excretion in patients with calcium stones [28]. The increased prevalence of metabolic syndrome over the past half century may partly explain the elevated prevalence of nephrolithiasis during that period [2,3,4].

Kidney stones can either be radiolucent or radiopaque [29]. Radiolucent stones include pure uric acid stones and indinavir-related stones [30]. Uric acid stones are the main radiolucent stones in the general population. Pure uric acid stones, which are characterized by their radiolucency on plain radiographs [31], can best be detected by computed tomography [32]. However, a combination of abdominal ultrasound and plain radiographs is a common cost-effective diagnostic alternative for radiolucent stones [31,32,33,34].

A Brazilian study reported a nine-fold increase in the risk of lithiasis among workers in high-temperature workplaces [35]. A case study from the southern Indian steel industry also showed an elevated risk of kidney stones among workers exposed to high occupational heat stress [17]. Taiwan is located in a subtropical (northern) to tropical (southern) area. We also have a high prevalence of nephrolithiasis of up to 9.6% [36]. In addition to the hot temperature in Southern Taiwan throughout the year (28 to 35 degrees Celsius), there is a high density of steel-related factories in which many workers need to be exposed to high-temperature environments. However, there are no related data about the prevalence of radiolucent stones among workers in high-temperature workplaces. The aim of this study was to investigate the risk of radiolucent stone formation among workers in high-temperature workplaces and the related risk factors associated with the condition.

## 2. Materials and Methods

The study protocols were reviewed and approved by the Institutional Review Board of Kaohsiung Medical University Hospital (KMUHIRB-E(II)-20210116, date of approval: 24 May 2021). We used the data of workers’ health examinations by the Taiwan Occupational Safety and Health Administration (TOHSA). Because the individual information was scrambled, written informed consent was waived by the institutional review board.

For the current study, the diagnosis of radiolucent nephrolithiasis was made according to a method described in the literature [34,37]. In brief, the plain abdominal radiographs of workers who were diagnosed with nephrolithiasis based on abdominal sonography were scrutinized to identify those with radiolucent stones to be included in the present investigation. On the other hand, workers with the findings of nephrolithiasis from plain radiographs were grouped under the category “other stones”.

The participants who fulfilled three or more of the following five criteria, published by the Bureau of Health Promotion, Department of Health, Executive Yuan in Taiwan, were diagnosed as having metabolic syndrome: (1) a waist circumference larger than or equal to 90 cm in men or 80 cm in women; (2) a medical history of hypertension, a systolic blood pressure higher than or equal to 130 mmHg, or a diastolic blood pressure higher than or equal to 85 mmHg; (3) a medical history of diabetes mellitus or a fasting serum glucose higher than or equal to 100 mg/dL; (4) a circulating high-density lipoprotein cholesterol (HDL) concentration lower than 40 mg/dL in men or 50 mg/dL in women; and (5) a serum triglyceride level higher than or equal to 150 mg/dL.

Six departments of the steel industry were grouped into hot-temperature workplaces, including the departments of the electric furnace, the converter, vacuum oxidization decarbonization, preparation, the continuous slab caster, and the continuous billet caster. The workers in these six departments were categorized as the group of “heat exposure”. The other workers were grouped as “non-heat exposure”.

All serum and urine samples were analyzed by a hospital with a central laboratory that was certified by an international certificate system performed by the Taiwan Accreditation Foundation (TAF). Serum creatinine and uric acid were measured with commercial kits using an automatic biochemical analyzer (SPOTCHEM EZ SP-4430) in accordance with the manufacturer’s instructions (Arkary, Inc., Kyoto, Japan). Midstream urine specimens were collected in a urine collection kit (Becton, Dickinson and Company, East Rutherford, NJ, USA) with a volume of 5–10 mL from all participants in the morning, considering the circadian rhymes of urine pH values under physiological conditions. Urinalysis was performed by an automatic urine analyzer system (Cobas^®^ 6500 urine analyzer, F. Hoffmann-La Roche Ltd., Basel, Switzerland).

We compared the three groups (radiolucent stones, other stones, and a non-stone group), using a descriptive analysis, an ANOVA (analysis of variance), and a chi-square test cross table. Then, we used logistic regressions to analyze the ORs (odds ratios) with 95% confident intervals to elucidate the relative risks of potential factors that may influence the incidence of radiolucent stone formation, including age, urine pH and gravity, and serum creatinine and uric acid concentrations as well as heat exposure. A linear regression analysis was conducted to determine the significance of the correlations between the factor(s) identified by the logistic regression and the probability of radiolucent stone formation. We performed the analyses using SPSS (IBM SPSS Statistics, IBM, Armonk, NY, USA) version 20 with a two-tailed α-level of 0.05.

## 3. Results

Of the 1681 steel workers enrolled in this study, 1480 (88%) showed no evidence of nephrolithiasis after abdominal sonographic and plain abdominal radiographic examinations. On the other hand, 116 workers (6.9%) exhibited nephrolithiasis on both examinations (i.e., the “other stone” group), and 85 (5.1%) workers were diagnosed as having radiolucent nephrolithiasis by demonstrating positive findings on abdominal sonographic examinations without supporting evidence from plain radiographs (Table 1). The average age of the radiolucent stone group was 37.9 years, and this group was significantly older than the non-stone group by 2.3 years. The group of other stones had the longest working duration among the three groups. Other factors such as urine pH, urine gravity, the serum uric acid level, and serum creatinine did not show significant differences among the three groups in the ANOVA tests. In addition, the workers with renal stones, including radiolucent stones, had a significantly higher prevalence of abnormal urinary RBC and WBC counts, according to the chi-square test.

The group with radiolucent stones was significantly older and had more heat exposure than the non-stone group. Table 2 shows that workers exposed to heat environments had a higher prevalence of radiolucent stones (18/206 vs. 67/1359, *p* = 0.025). The group with radiolucent stones had 61.2% of workers older than 35 years, while this number was 46.1% in the non-stone group (*p* = 0.007). Most of the workers in this study were males, but no significant difference in gender distribution was noted between the two groups.

Table 3 shows the results of a logistic regression of radiolucent stones predicted by age, working duration, metabolic syndrome, gender, heat exposure, urine pH, urine gravity, serum uric acid level, and creatinine. Although our results identified age, heat exposure, and hyperuricemia as risk factors for nephrolithiasis, only age demonstrated statistical significance, with an odds ratio of 1.052 (*p* = 0.008). On the other hand, despite the apparently higher odds ratio for heat exposure (OR = 1.728), it missed statistical significance by a narrow margin (*p* = 0.052). Because this was a cross-section study, the abnormal urinary RBC and WBC counts may have been induced by renal stones. As a result, we only included urine pH, urine gravity, serum uric acid, and serum creatinine in our logistic regression analysis.

To elucidate the association between age and heat exposure, we regrouped our subjects by age with 35 years being adopted as the cut-off point, taking into consideration their mean age of 35.73 years and median age of 35 years before investigating the effect of heat exposure. Four subgroups were obtained after regrouping by age and heat exposure, namely those with an age ≤35 years without heat exposure (group 1, n = 726); those with an age >35 years without heat exposure (group 2, n = 633); those with an age ≤35 years with heat exposure (group 3, n = 105); and those with an age >35 years with heat exposure (group 4, n = 101). The *p*-value of the chi-square test was 0.516, indicating no difference in the age distribution between the groups with and without heat exposure.

Through grouping by age and heat exposure, a logistic regression of the significance of potential predictors of radiolucent stones, including working duration, metabolic syndrome, gender, urine pH, urine gravity, serum uric acid level, and serum creatinine demonstrated the statistical significance of all three groups, with odds ratios of 2.126, 2.695, and 2.899 for groups 2, 3, and 4, respectively, when compared with group 1 (Table 4).

We used the probability of radiolucent nephrolithiasis calculated from the last model to evaluate the impacts of age and heat exposure, as shown in Figure 1. We found a lack of association between the probability of nephrolithiasis and age among workers with heat exposure (r^2^ = 0.002, *p* = 0.504), while the probability of radiolucent was nephrolithiasis significantly correlated with age in workers without heat exposure (r^2^ = 0.323, *p* < 0.001).

## 4. Discussion

Our research found an increased prevalence of radiolucent stones among workers exposed to high temperatures, regardless of their age, while those without heat exposure showed an increased prevalence with age. Our findings were consistent with those of previous studies [37,38,39]. The prevalences of nephrolithiasis and radiolucent stones in this study were 12.0% and 5.1%, respectively. The overall prevalence of nephrolithiasis in Taiwan is 9.6%, with an age-adjusted prevalence of 9.01% in males [36,40]. The prevalence of nephrolithiasis was higher in steel workers than the general population. The possible mechanism of heat-exposure-induced radiolucent nephrolithiasis may be related to chronic dehydration due too much sweating in a high-temperature workplace [38,41]. Thus, reminding workers under heat exposure to increase water and electrolyte intake can not only avoid heat injuries such as heat stroke and heat exhaustion but can also improve dehydration and decrease the risk of nephrolithiasis.

Although the serum uric acid level in the group with radiolucent nephrolithiasis was higher than that in the non-stone group, with the odds ratio of serum uric acid being 1.184-1.185, despite the absence of statistical significance (*p*-values were 0.054 on first logistic regression and 0.055 on second logistic regression). In addition, urine pH and serum creatinine were not significantly different between the two groups, suggesting relative minor impacts of urological factors on radiolucent nephrolithiasis compared to heat exposure and age.

Although heat exposure, age, and gender are well-known risk factors for nephrolithiasis [35,37,39,42], the results of the current study suggest an interaction among these factors by identifying age as a risk factor for radiolucent nephrolithiasis without heat exposure. The odds ratio of age was 1.052 in the first logistic regression model, while it was 2.126 for group 2 versus group 1 in the second logistic regression model. The findings implied that the risk of radiolucent nephrolithiasis among workers aged >35 years was more than double compared to that in their younger counterparts in the absence heat exposure, suggesting that age was not only a risk factor for general nephrolithiasis but also a risk factor for radiolucent stone disease. Interestingly, while we found an increased probability of radiolucent stone formation with age among workers without heat exposure, the probability remained high in workers with heat exposure regardless of their age. Despite the previous finding that uric acid stone and atypical stones were associated with the older population, the current study showed that younger individuals may also be at risk of radiolucent stone formation (e.g., uric acid stone) if their occupation involves heat exposure. In this study, the lack of a significant association between gender and the prevalence of radiolucent stones may partly be attributed to the notably lower proportion of females compared to males, which may obscure the significance of this finding.

Although a number of previous studies showed a correlation between metabolic syndrome and uric acid nephrolithiasis, we did not observe a significant association in our study. The average age of the workers in the factory was 35.7 years, which may explain the low prevalence of metabolic syndrome that usually develops in individuals with advanced age. Such a low prevalence of metabolic syndrome in our study population (i.e., 18.1%) may have contributed to the lack of a significant correlation with radiolucent nephrolithiasis.

Moreover, urine pH has been found to affect the risk of urolithiasis; while a pH below 5.5 is known to increase the risk of uric acid stone formation, a pH over 6.0 favors the development of calcium stones. In view of such an impact, we included urine pH in our logistic regression analysis, which demonstrated no significant impact of pH on the incidence of radiolucent nephrolithiasis in our participants.

We defined radiolucent stones as positive abdominal ultrasound findings without radiopacities on plain radiography. A prior study suggested the effectiveness of sonography as a diagnostic tool for detecting kidney stones in patients with suspected nephrolithiasis [33]. Consistently, a previous clinical study on 62 patients showed that both computed tomography and ultrasound were excellent for diagnosing ureteral stones [34]. Considering the cost-effectiveness and radiation exposure, sonography appeared to be a reasonable first choice in the diagnosis of urolithiasis. Due to the radiolucent characteristic of pure uric acid stones, ultrasound examination of the kidneys should always be considered if uric acid stones are suspected [31]. Despite the common acceptance of dual-energy computed tomography as the gold standard for the direct visualization of uric acid stones [32], the associated cost remains an important concern. Therefore, we suggest the combined use of plain abdominal radiography with sonography for the diagnosis of radiolucent stones, taking into consideration the cost of radiation exposure and the risk of allergy to the contrast medium pertinent to computed tomography.

Because we did not have information about the volume of water drank and the diet content of the participants in this study, the potential contributions of hydration and diet-related factors to urolithiasis remain unclear. Nevertheless, given that the company did not have any policy to restrict its workers’ water intake or their use of the toilet, the study subjects were supposed to be well hydrated. In addition, the provision of similar food for lunch by the company for its workers, regardless of the departments in which they worked (i.e., with or without heat exposure), may also preclude a significant influence of food on nephrolithiasis. On the other hand, although meat consumption is a main risk factor for uric acid stone formation, we did not include relevant items in the questionnaire to acquire such information. Finally, heterogeneity in examination results may bias our findings because the abdominal sonographic studies were not conducted by a single gastroenterologist, and the reports of plain abdominal radiographs were produced by multiple radiologists due to the large number of participants in the current study. However, provided that all gastroenterologists and radiologists involved in the present investigation were board-certified and well qualified for their assignments, it was a rational assumption that the impact of inter-rater variation on the interpretation of examination findings was minimal.

## 5. Conclusions

The current study involving over a thousand steel workers demonstrated a significant positive association between heat exposure and radiolucent nephrolithiasis without notable impacts from gender, urine pH and specific gravity, and serum creatinine and uric acid concentrations. In contrast to an increased incidence of nephrolithiasis with age among those without heat exposure, an elevated incidence was noted in heat-exposed individuals regardless of their age. The finding underscored a potential increase in the risk of radiolucent nephrolithiasis in the relatively young population who work in an environment involving constant heat exposure.

## Figures and Tables

**Figure 1 ijerph-19-15720-f001:**
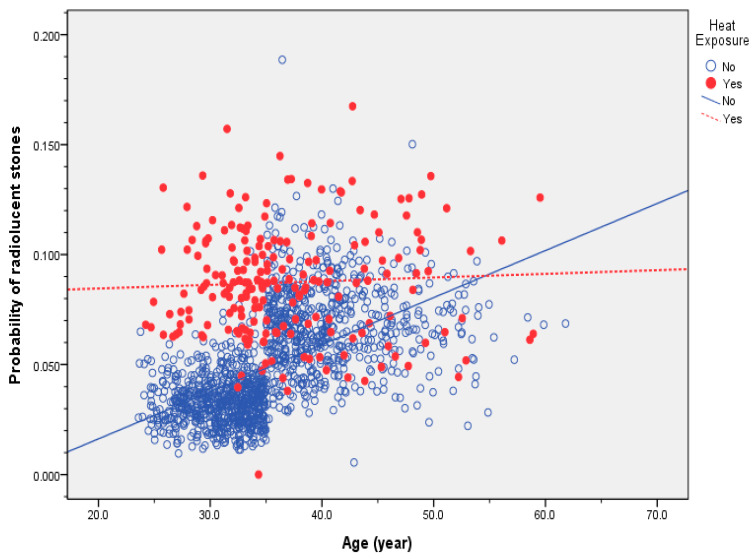
Correlations between the probability of radiolucent stones and age in steel workers with and without heat exposure.

**Table 1 ijerph-19-15720-t001:** Frequency distribution and ANOVA tests of the non-stone, radiolucent stone, and other stone groups by age, work years, urine pH, urine gravity, urinary WBC, urinary RBC, serum uric acid, and creatinine.

	Non-Stone n (%) or Mean (S.D.)	Radiolucent Stone n (%) or Mean (S.D.)	Other Stone n (%) or Mean (S.D.)	*p*-Value *
Number	1480 (88.0%)	85 (5.1%)	116 (6.9%)	
Male/Female	1395/85	83/2	113/3	0.157
Age (years)	35.6 (6.4)	37.9 (6.7)	37.7 (6.7)	<0.001
Working duration (years)	8.5 (4.2)	9.3 (4.4)	9.7 (3.5)	0.003
Urine pH	6.35 (0.76)	6.37 (0.81)	6.46 (0.77)	0.391
Urine gravity	1.0210 (0.0068)	1.0207 (0.0057)	1.0196 (0.0068)	0.087
Urine WBC > 5/HPF	56 (3.8%)	6 (7.1%)	11 (9.5%)	0.07
Urine RBC > 5/HPF	52 (3.5%)	4 (4.7%)	10 (8.6%)	0.024
Serum uric acid (mg/dL)	6.85 (1.44)	7.06 (1.37)	6.99 (1.50)	0.279
Serum creatinine (mg/dL)	1.22 (0.38)	1.20 (0.13)	1.23 (0.17)	0.870

* Chi-square test for categorical data and ANOVA (analysis of variance) for continuous data: S.D.: standard deviation; WBC: white blood cell; RBC: red blood cell.

**Table 2 ijerph-19-15720-t002:** Chi-square test of radiolucent stone and non-stone groups by gender, heat exposure, and age.

	Non-Stone	Radiolucent Stone	*p*-Value *
Heat exposure			0.025
No (%) Yes (%)	1292 (87.3%) 188 (12.7%)	67 (78.8%) 18 (21.2%)	
Age			0.007
≤35 years (%) >35 years (%)	798 (53.9%) 682 (46.1%)	33 (38.8%) 52 (61.2%)	
Gender			0.185
Female (%) Male (%)	85 (5.7%) 1395 (94.3)	2 (2.4%) 83 (97.6%)	
Metabolic syndrome			0.322
No (%) Yes (%)	1208 (81.6%) 272 (18.4%)	73 (85.9%) 12 (14.1%)	

* Chi-square test.

**Table 3 ijerph-19-15720-t003:** Logistic regression of radiolucent stones predicted by age, working duration, metabolic syndrome, gender, heat exposure, urine pH, urine gravity, serum uric acid, and creatinine.

	Odds Ratio	95% Confidence Interval of Odds Ratio		*p*-Value
Heat exposure (yes vs. no)	1.728	0.995	3.001	0.052
Age (years)	1.052	1.013	1.092	0.008
Working duration (years)	1.005	0.942	1.072	0.887
Metabolic syndrome (yes vs. no)	0.590	0.310	1.122	0.108
Gender (male vs. female)	2.243	0.485	10.375	0.301
Urine pH ^1^	1.019	0.761	1.364	0.901
Urine gravity	0.147	<0.001	>1000	0.910
Serum uric acid (mg/dL)	1.185	0.997	1.408	0.054
Serum creatinine (mg/dL)	0.173	0.026	1.150	0.069

^1^ Abbreviation: pH (potential of hydrogen).

**Table 4 ijerph-19-15720-t004:** Logistic regression of radiolucent stones predicted by working duration, metabolic syndrome, gender, urine pH, urine gravity, serum uric acid, and serum creatinine following grouping by age and heat exposure.

	Odds Ratio	95% Confidence Interval of Odds Ratio		*p*-Value
Group 2 vs. Group 1 ^1^	2.126	1.163	3.884	0.014
Group 3 vs. Group 1	2.695	1.201	6.049	0.016
Group 4 vs. Group 1	2.899	1.235	6.805	0.015
Working duration (years)	1.010	0.947	1.077	0.765
Metabolic syndrome (yes vs. no)	0.603	0.317	1.149	0.124
Gender (male vs. female)	2.264	0.488	10.507	0.297
Urine pH ^2^	1.005	0.752	1.344	0.972
Urine gravity	0.036	<0.001	>1000	0.843
Serum uric acid (mg/dL)	1.184	0.996	1.407	0.055
Serum creatinine (mg/dL)	0.172	0.025	1.160	0.071

^1^ Group 1: age ≤ 35 years without heat exposure; Group 2: age > 35 years without heat exposure; Group 3: age ≤ 35 years with heat exposure; Group 4: age > 35 years with heat exposure. ^2^ Abbreviation: pH (potential of hydrogen).

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
