# Peer review of "The Risk Factors for Radiolucent Nephrolithiasis among Workers in High-Temperature Workplaces in the Steel Industry"

_ijerph, 2022, doi:10.3390/ijerph192315720_

Round 1

Reviewer 1 Report

1. Page 6, Line 177, double typed of "would increase" needs to be removed.

2. Is there any information regarding the urinary routine tests (e.g. urinary RBC) to compare with both (or between) the radiolucent or non-radiolucent Nephrolithiasis ?  I would assume that the result of the urine routine tests should be available since it is easily abstracted from a physical check up.  My curiosity might be satisfied if the authors could provide some idea on this matter, however, if it has not been checked, I would not mind. 

Reviewer 2 Report

The paper is in the scope of “International Journal of Environmental Research and Public Health”. The publication can be reconsider after the improvements as per the following comments.  Author should to work on the following comments:

1.      What is the novelty of this research work?

2.      Abstract section: What is the meaning of last line in the abstract section (Line number: 34-35)?

3.      Materials and Methodology: Author should have to mention the measurement techniques, data collection strategies and model number of used instruments. The information is not clear in this section. Author can add any tabular form to justify this section.

4.      Urine pH is one of the important factor to study the circumstances. Why author have not mention any information of urine pH?

5.      In Results Section: Figure 1, author should have to add more discussion for this figure.

6.      Discussion Section: In most of the cases, author discussed and given the references of most of the previous studies. Author should have to discuss the concept based on his/her own research based data.

7.      Conclusion Section: This section of the paper is very weak. Author given certain recommendation to prevent the radiolucent nephrolithiasis. Author should have to mention qualitative points in this section.

8.      There are many linguistics mistakes in the manuscript. Author should have to rectify the mistakes of commas and connectors.

Minor Comments

9.      Line number 177: “Would increase” word is repeated in the sentence.

10.  Line number: 207 – 208: Sentence mistake. Author should have to re-write this line.

11.  Line number 213, 218: Author should have to write the authors names as per journal guidelines.

Reviewer 3 Report

The present work provides relevant information regarding the risk of radiolucent stone formation among workers at a high temperature workplace and the related risk factors associated with the condition.

In my opinion, this research topic is of paramount importance and the paper is well written and the results are well discussed and organized. Consequently, I strongly recommend its publication in the International Journal of Environmental Research and Public Health.

Minor observations:

Page 5, lines 167 and 169 – Please put the square of the correlation coefficient in superscript.

Page 6, line 177 – Please delete the duplicate entry: “would increase”.

Page 6, line 198 – “…Group 2 versus. Group 1… Please delete the point after “versus”.

Page 6, line 225 – “… the volume of water drinking and the diet…”. Please correct the sentence; I would suggest replacing “drinking” by “drunk”.

Reviewer 4 Report

The paper presents the following main issues that in my opinion hinder its publication:

1. Meat consumption is a main risk factor for uric acid stones that needs to be addressed

2. Heat exposure is an arbitrary variable

3. Logistic regression investigating interaction between age and heat exposure would have been preferable to the creation  of 4 subgroups according to both dichotomised variables. 

Round 2

Reviewer 4 Report

The text in the lines 124-132 needs careful english editing
